# Complementary Therapy Learning in the Setting of Lung Transplantation: A Single-Center Observational Study of Appropriation and Efficacy

**DOI:** 10.3390/jcm12051722

**Published:** 2023-02-21

**Authors:** Mireille Michel-Cherqui, Julien Fessler, Barbara Szekely, Matthieu Glorion, Edouard Sage, Marc Fischler, Alexandre Vallée, Morgan Le Guen

**Affiliations:** 1Department of Anesthesiology and Pain Management Clinic, Hospital Foch, 92150 Suresnes, France; 2Department of Thoracic Surgery and Lung Transplantation, Hospital Foch, 92150 Suresnes, France; 3School of Medicine, UFR Simone Veil, University Versailles Saint-Quentin-en-Yvelines (Paris Saclay University), 78000 Versailles, France; 4Department of Epidemiology-Data-Biostatistics, Delegation of Clinical Research and Innovation, Hospital Foch, 92150 Suresnes, France

**Keywords:** lung transplantation, self-hypnosis, relaxation, perioperative pain, anxiety

## Abstract

Transplanted patients could benefit from complementary techniques. This prospective single-center, open study, performed in a tertiary university hospital, evaluates the appropriation and efficacy of a toolbox-kit of complementary techniques. Self-hypnosis, sophrology, relaxation, holistic gymnastics, and transcutaneous electric nerve stimulation (TENS) were taught to adult patients scheduled for double-lung transplantation. Patients were asked to use them before and after transplantation, as needed. The primary outcome was appropriation of each technique within the first three postoperative months. Secondary outcomes included efficacy on pain, anxiety, stress, sleep, and quality-of-life. Among the 80 patients included from May 2017 to September 2020, 59 were evaluated at the 4th postoperative month. Over the 4359 sessions performed, the most frequent technique used before surgery was relaxation. After transplantation, the techniques most frequently used were relaxation and TENS. TENS was the best technique in terms of autonomy, usability, adaptation, and compliance. Self-appropriation of relaxation was the easiest, while self-appropriation of holistic gymnastics was difficult but appreciated by patients. In conclusion: the appropriation by patients of complementary therapies such as mind–body therapies, TENS and holistic gymnastics is feasible in lung transplantation. Even after a short training session, patients regularly practiced these therapies, mainly TENS and relaxation.

## 1. Introduction

Pain, anxiety, and sleep disorders are common in patients undergoing lung transplantation. Each stage of the process can place the patients in extreme situations, and be sources of pain, discomfort, and stress.

Severe alteration of quality of life has been reported in patients awaiting lung transplantation [1,2]. In our experience, at inscription on the waiting list, 59 % of patients report pain and 36% report significant anxiety [3]. After transplantation, prevalence of 6-month postoperative pain is about 50% [4,5,6] and can reach 68%, as recently reported [7,8]. Furthermore, numerous studies have reported poor quality of life in patients with pain [8]. In these patients, active coping, characterized by an active engagement in dealing with stressful events is associated with better quality of life [9]. Consequently, this regular practice involving active patient participation needs to be developed. On the other hand, many complementary techniques have proved their efficacy in the management of chronic and acute pain, anxiety, and sleep disorders. Among complementary techniques we decided to study non-pharmacological interventions which could be taught easily and practiced regularly by the patients throughout the experience of lung transplantation.

Mind–body techniques, like relaxation, mindfulness, and hypnosis, are used for pain and anxiety management in perioperative care. However, the quality of evidence for its efficacy in improving post-surgical outcomes is limited as reported in a systematic review [10]. Their place in solid organ transplantation has begun to be studied with favorable results in kidney, kidney/pancreas, liver, heart, or lung transplanted patients [11]. They have been more recently proposed for patients waiting for kidney transplantation [12,13]. In patients awaiting lung transplantation, higher levels of coping and lower levels of disengagement have been shown to be associated with better psychological quality of life [9]. Interestingly, telephone-based coping skills training allowed significant improvements in quality of life in these patients [14]. Use of transcutaneous electric nerve stimulation (TENS) allows a decrease in pain killers in patients with pain [15,16] and physical training, which has been part of prehabilitation programs before surgery, allows improvement of the postoperative recovery course [17].

The interest of implementing these techniques before lung transplantation has never been studied. We wanted to take advantage of the evaluation period in our transplant unit before enrolment on the waiting list to propose one training session of the different techniques and to encourage patients to practice regularly. We hypothesized that patients could learn and practice these techniques during evaluation and preparation for lung transplantation. This could be integrated into regular practice and used, as necessary, as a tool. We also hypothesized that this practice could improve patients’ experience, especially regarding pain and quality of life. The purpose of the study was to evaluate the implementation of these techniques and to assess the effectiveness of these therapies.

## 2. Materials and Methods

### 2.1. Ethics Approval

This study was performed in accordance with the Declaration of Helsinki. Ethical approval was provided by the Ethical Committee Ile-de-France VII (reference N° SC 16-035; 2 February 2017; Chairperson F. Boissy, Paris, France). The study was designed in accordance with the STROBE Statement and the protocol was published on the Clinical.trials.gov website (NCT03167528; 30 May 2017). Our Lung Transplant group attests that they performed all procedures in strict compliance with the International Society for Heart and Lung Transplantation ethics statement. The complete protocol, registered with the competent authorities under the N° ID-RCB 2016-A01654-47, can be obtained on request. Patients and the public were not involved in the design, conduct, reporting, or dissemination plans of the research.

### 2.2. Study Design, and Setting

This institutionally sponsored, open, single-center study was performed in a tertiary care university hospital.

### 2.3. Patient Population

Adult patient candidates for a double-lung transplantation undergoing pre-transplant evaluation were informed of the study at the anesthetic consultation and they were included after their registration on the waiting list. Written informed consent was obtained from all patients.

### 2.4. Study Protocol—Techniques

The opportunity to learn each technique was proposed to the participants. The global process of the study was explained to the patients, including the description of the different techniques, their usefulness, and the necessity to practice them regularly before and after transplantation. A booklet recapitulated the information. A description of the various techniques (relaxation, self-hypnosis, sophrology, and holistic gymnastics) is presented in a Appendix A. In addition, the possibility of using TENS was mentioned to each patient but it was taught before transplantation only to patients describing pain where TENS could be useful. TENS was taught after transplantation to each patient presenting pain poorly relieved by epidural or usual antalgics. The technique was taught by pain nurses in charge of the study.

Repeated training sessions were possible pre transplantation in patients staying for a long time or repeatedly in the hospital before the transplantation, after the transplant, additional sessions were performed at the request of the patients and/or the team. This disparity is possibly a limiting factor in this study.

### 2.5. Study Protocol—Anesthesia, Surgery, and Postoperative Analgesia

Anesthesia technique includes epidural analgesia inserted before surgery and used during it. The complete anesthetic protocol is described on the website http://anesthesie-foch.org/protocoles-anesthesie/, accessed on 3 January 2023 (section Anesthesia Foch lung transplant protocol).

Surgery was performed via two successive antero-lateral incisions.

During the acute postoperative phase, epidural analgesia was continued for five days, monitored twice daily by pain nurses, and relayed by oral opioids. Multimodal analgesia was started intraoperatively and continued during the acute phase.

### 2.6. Data Collection (Figure 1)

Pain was assessed by (i) the answer to the question “do you think you regularly suffer from pain?”; (ii) self-assessment using a visual analogue scale between 0 (no pain) and 100 (worst pain imaginable) at rest during the consultation, as the maximum pain intensity during the last three days, and as usual pain during the last three days; (iii) the response to the DN4 questionnaire (neuropathic nature of pain) [18]. Thoracic location of pain and its relation to care was noted. Analgesic treatment, including treatment of neuropathic pain, non-medicinal treatment, and benzodiazepine, was specified.Anxiety and depression were assessed with the French version of the Hospital Anxiety and Depression Scale (HADS); 14 items rated from 0 to 3, seven questions related to anxiety and seven others to depression; maximum of each score = 21. An anxiety score or a depression score above 7 on this 14-item scale was considered as indicative of a clinical diagnosis of anxiety or depression [19].Perceived stress was evaluated with the perceived stress score (ten items rated from 0 to 5; total score < 21 = someone who knows how to manage stress; Total score between 21 and 26 = someone who usually knows how to manage stress, except in certain situations; total score > 27 = someone who does not know how to manage stress) [20].Sleep quality was assessed by the Spiegel sleep score (six items rated from 0 to 5: the lower the score, the more serious the sleep disorder [21].Quality of life was assessed by the EuroQol visual analog scale (EQ-VAS) which records the respondent’s self-rated health on a vertical scale graduated from 0 to 100 between the two extremes “the best health you can imagine” and “the worst health you can imagine” [22].The number of sessions of complementary techniques before and after transplantation was recorded and separated into sessions by the patient, completed by the patient alone or with a therapist (training sessions).Patients’ appropriation of the techniques was evaluated by the patients and caregivers using a specific questionnaire. The qualities of the techniques, the patients’ implication and degree of satisfaction were measured by an 11-point numerical rating scale with 0 being “worst possible” and 10 “excellent” on five features (usability, conformity, autonomy, adaptation, and involvement). Patients evaluated their impression of benefit using the same scale.Pain-related impairment of daily activities after thoracic surgery was assessed with a specific questionnaire [23] using its French translation [24]. Using this questionnaire, patients scored their pain impairment for each activity from 0 (“no pain or activity never performed”) to 1 (“pain impairs me a little to perform this activity”), 2 (“pain somewhat impairs me”), 3 (“pain impairs me a lot”) or 4 (“pain prevents me from performing this activity”). Activities are separated into daily or routine activities (getting out of bed for example) and chosen or task activities (swimming or carrying heavy bags for example).

**Figure 1 jcm-12-01722-f001:**
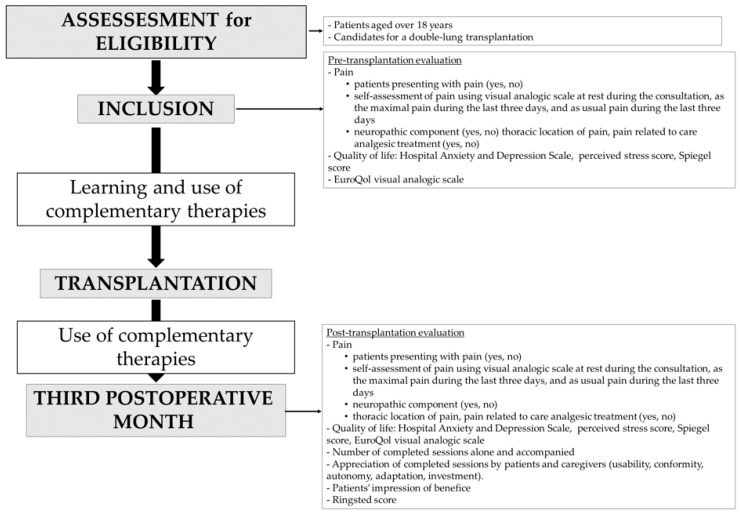
Summarizes the data collected at each time-point.

### 2.7. Primary and Secondary Outcomes

The primary outcome of the study was the compliance of patients or their self-appropriation regarding each technique evaluated by the number of sessions of complementary techniques practiced by the patient alone, before transplantation and up to three months after.

The first secondary outcome was the evaluation by the patients and by their caregivers of the implication and satisfaction regarding the different techniques using five features: usability, conformity to instructions, autonomy, adaptation to different situations, and involvement.

Other secondary outcomes were the efficacy of complementary techniques on pain and on quality of life (anxiety, depression level, stress level, sleep, and global evaluation).

### 2.8. Sample Size Calculation

As little is known concerning this topic, the number of patients was primarily determined by the number of transplants performed in a year (65) plus the number of patients potentially evaluated but not transplanted (5) and the number of patients who would have died between inclusion and the third month after transplantation (10). Inclusion was stopped when 80 patients were included.

### 2.9. Statistical Analyses

Patients were included in the analysis if they learnt at least one mind-body technique (i.e., sophrology, relaxation, self-hypnosis), and if they were transplanted within a year after the pre-transplant assessment.

All statistical analyses were done on an intention-to-treat basis. There was no imputation of missing data.

Categorical variables are presented as numbers (proportion) and compared between groups using the McNemar test; continuous variables are presented as median (interquartile range) and compared between groups using the Wilcoxon signed-rank test.

Comparison of the usability, conformity to instructions, autonomy, adaptation to different situations, and involvement between all complementary therapies used the Kruskal–Wallis test with Dunn’s multiple comparisons test in case of statistical significance. The Pearson correlation measured the strength of the linear relationship between the number of performed sessions and indices of quality of life.

All tests were two-sided. *p* values of less than 0.05 were considered significant. The statistics were generated using SAS 9.4 software.

## 3. Results

The study took place from May 2017 (beginning of inclusions) to September 2020 (last evaluation), 84 patients were approached for participation. Eighty patients gave their consent to the study and were included, 64 were transplanted and 59 were evaluated three months after transplantation (Figure 2).

### 3.1. Patients’ Characteristics at Inclusion

Patients’ characteristics at inclusion are described in Table 1. Most patients had cystic fibrosis or emphysema or chronic obstructive pulmonary disease. Eight patients (40%) had pain at inclusion, but pain was moderate, and most patients did not receive painkillers. They had high perceived stress and poor quality of life.

### 3.2. Performed Sessions

Regarding the pretransplant training sessions, all patients had at least one mind–body training session. Relaxation was taught to 53 patients, holistic gymnastics, and self-hypnosis to 44 patients and sophrology to four patients. Among the patients with preoperative pain, TENS was proposed to eight patients.

Regarding the post-transplant period, 16 patients had relaxation training sessions, 28 self-hypnosis and only one had sophrology. Eighteen patients had holistic gymnastics training sessions. TENS was taught to 35 patients.

A total of 4359 sessions were performed. Details of the repartition of training and self-practice sessions and pre- and post-transplantation sessions are shown in Table 2. The most frequently used techniques were relaxation before transplantation and relaxation and TENS after transplantation.

The number of sessions performed by each patient before and after transplantation is reported in Figure 3.

### 3.3. Patients’ and Caregivers’ Assessments of the Techniques

Patients’ and caregivers’ assessments of the techniques at the end of the study are reported as a radar presentation. It summarizes their assessments according to the five studied features (Figure 4), details are available in Table 3.

Use of TENS was the most efficient in terms of autonomy, but also usability, adaptation, and conformity. Mind–body techniques were globally judged appreciable, among them, self-appropriation of relaxation appeared to be the easiest. Self-appropriation of holistic gymnastics was difficult, but this technique was appreciated by the patients.

### 3.4. Patients’ Characteristics Three Months after Transplantation

Patients’ characteristics three months after transplantation are presented in Table 1. The percentage of patients with pain was close to that observed at inclusion (40.3% at inclusion and 35.6% at three months evaluation; *p* = 0.532) but with a significant increase in the proportion of patients with neuropathic pain (3.4% at inclusion and 27.6% at three months; *p* = 0.0008). There was a global improvement of almost all the quality-of-life variables. Anxiety and depression were greatly improved (*p* < 0.0001 for both variables). Perceived stress score was also better after transplantation (*p* < 0.0001) and the global quality of life assessed by the EQ-VAS was greatly improved (*p* < 0.0001). This was confirmed by low indexes obtained in the Ringsted quality of life questionnaire. On the other hand, sleep remained of poor quality.

### 3.5. Relation between the Number of Performed Sessions and Indices of Quality of Life

There was no statistical relationship between the total number of performed sessions and Ringsted total score and absolute value of EQ-VAS measured three months after transplantation and change of EQ-VAS value recorded at inclusion and at third postoperative month (Appendix A).

## 4. Discussion

Our study shows that implementation of complementary practices is feasible in lung transplantation. Patients and caregivers globally reported the patient’s appropriation of these techniques. Although it failed to conclude that these practices improve pain, anxiety and quality of life, most patients reported considerable benefit. The large number of sessions practiced was in favor of a strong adhesion of the patients to some of these techniques. The purpose was to offer tools to patients, and these were largely used.

Mind-body techniques include a large array of practices and have already been studied in perioperative care, especially to address emotional distress. In many studies including large populations, pre-surgery levels of emotional distress significantly predicted postoperative pain [25]. Among the mind–body techniques, relaxation could be very easy to teach. Relaxation was associated with decreased muscular tension and sympathetic nervous system activity [26], lowering cortisol levels and inflammatory processes, especially in long term practice [27]. However, it failed to show better pain control [28]. Hypnotic interventions have been shown to deliver meaningful pain relief [29]. In patients undergoing surgery, meta-analysis showed that hypnosis was effective in reducing emotional distress, pain, medication consumption and recovery [30,31].

Sophrology is a mind-body discipline close to relaxation and mindfulness. This is a structured method, well-known in French-speaking European countries, consisting of practical physical and mental exercises, using techniques such as concentration, deep breathing, relaxation, visualization, and simple movements which can have a positive role in reducing anxiety [32].

Although all these techniques have been largely evaluated, their implementation and self-appropriation by patients has been insufficiently studied. Two randomized studies have shown that practice of self-hypnosis prior to elective surgery reduced post-operative anxiety and resulted in less postoperative pain [33,34]. We recently reported our experience with self-hypnosis, showing that implementation of self-hypnosis was possible, but failing to demonstrate an improvement in patients’ experience, perhaps due to the variable compliance with the technique in a non-elective procedure [35].

Coping-skills training and a mindfulness-based stress reduction program have been proven to reduce anxiety, depression, poor sleep and to improve the quality of life in solid organ transplant recipients while waiting for transplantation [11,14].

Holistic gymnastics is based on smooth physical training and improvement of the body consciousness. It has never been evaluated in a perioperative setting, but physical training has been part of prehabilitation programs before surgery and has been shown to improve the postoperative recovery course [17]. Exercises could be home based and have been shown to reduce distress and improve depressive syndrome [36].

TENS is considered as an adjunct to core treatment for immediate short-term relief of pain, irrespective of diagnosis. It allows patients to tailor their treatment according to their individual needs [37]. In thoracic surgery, it has proved its efficacy as a part of multimodal analgesia [38]. Our study showed that appropriation of this technique was excellent. It had the highest rate of usefulness, autonomy, and benefit.

Finally, pain management after lung transplantation is complex. In addition to the fact that double-lung transplantation involves two thoracotomies, with their own risk of chronic pain, patients also suffer from pain related to care and to various postoperative complications, … The functional result also plays a role when it is not “perfect” influencing the psychological state.

### Weaknesses and Strengths of the Study

First, the main weakness of the study is the small number of patients able to learn all the techniques. Only 75% of the patients had self-hypnosis and four patients had sophrology. This was due mainly to the variable availability of the facilitators. Consequently, external validity is limited.

One of the goals of the study was to prove that, despite a small number of teaching sessions, patients could learn and use these techniques as a tool. This goal was clearly obtained for two techniques: relaxation and TENS. It encourages us and other teams to teach them routinely. Relaxation could be at least explained and practiced during a 10-min session during the pre-transplant assessment session. Teaching the use of TENS should at least be part of the post-operative care. Sophrology, hypnosis and holistic gymnastics were less practiced, they probably require more training.

Appropriation of techniques is also a good coping strategy to improve quality of life [9]. As appropriation of a technique seems to be poorly studied, we designed a questionnaire to evaluate it. Among the five qualities that we chose to evaluate, autonomy and usefulness seemed to be very discriminant among techniques. This type of questionnaire could be useful for other teams to evaluate the implementation and the appropriation of techniques in other settings.

Integrative medicine proposing a coordinated approach to conventional medicine and complementary treatments has become accepted in many fields like pain, oncology, or pediatrics [39] and is starting to be accepted in perioperative care [40]. Our work is among the first to propose an integrative approach for improving quality of life in lung transplantation.

A randomized study could perhaps have demonstrated the superiority of the implementation of these techniques. Likewise, in our previous study of self-hypnosis, patients were aware of these techniques thanks to social media communication, and they expected that this toolbox-kit could help throughout the transplantation process. We consequently decided that it would not be ethical to randomize this study.

A qualitative study would have been necessary to evaluate more accurately the real appropriation of the techniques. Many patients told us that practicing relaxation, sophrology or hypnosis had been a very efficient technique to focus on something else during an uncomfortable situation (non-invasive ventilation and fiberoptic bronchoscopy for example), to breathe more easily, especially prior to transplantation, and to help with sleep.

Our study design, with very different therapies from each other, can obviously be criticized but all techniques are routinely proposed by our Pain Management Clinic and were known by the lung transplantation candidates, and it was difficult for us to select only some of them.

Our relaxation and hypnotic procedure could also be discussed. Our sessions were not standardized using a script as proposed for many clinical trials [41], but we regularly compared and harmonized our practices with a monthly meeting. Sophrology and holistic gymnastics teaching were each performed by a unique therapist.

The design of our study did not include following up the patients during the pretransplant period and the assessment of their learning. They could have benefited from better support during the waiting period before transplantation by adding regular telephone calls [14]. Remote training was not part of our usual care. Unfortunately, COVID-19 reminded us that we should promote it as part of our future preoperative program as proposed by Blumenthal et al. [36].

## 5. Conclusions

Implementation and patients’ appropriation of complementary therapy like mind-body therapies, TENS and holistic gymnastics is feasible in lung transplantation. Even after one training session, patients regularly practice these therapies, essentially TENS and relaxation. Remote training should be associated with teaching. Further studies, especially qualitative ones, should allow us to define the role of complementary therapies in improving patients’ quality of life, and promote an integrative approach to lung transplantation. The question may arise about introducing such a program as soon as the respiratory situation deteriorates and the indication for transplantation is considered. Indeed, the time between registration on the waiting list and lung transplantation is usually a few weeks, which leaves little time for learning and practicing the various techniques. In addition, an earlier start would prevent the learning of techniques from being influenced by the availability of teachers.

## Figures and Tables

**Figure 2 jcm-12-01722-f002:**
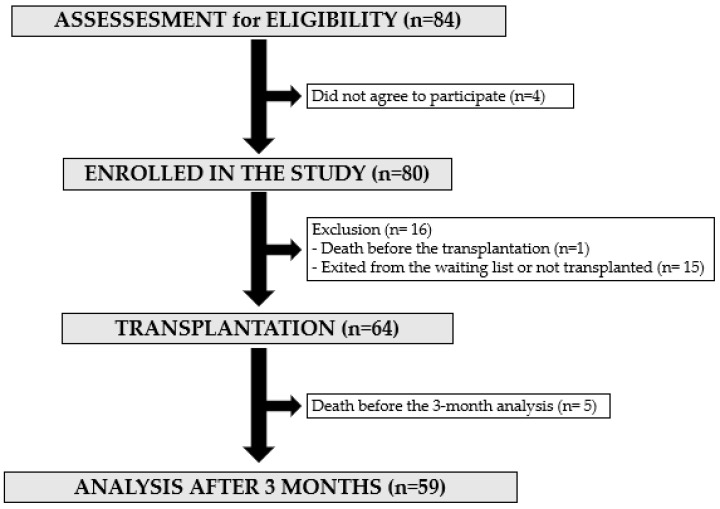
Flow chart.

**Figure 3 jcm-12-01722-f003:**
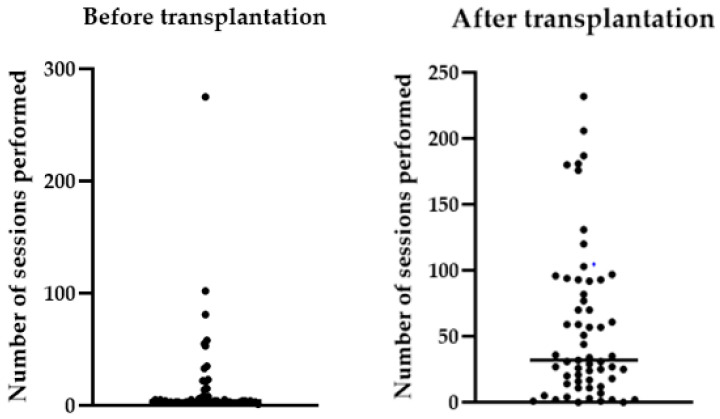
Number of sessions performed by each patient before and after transplantation.

**Figure 4 jcm-12-01722-f004:**
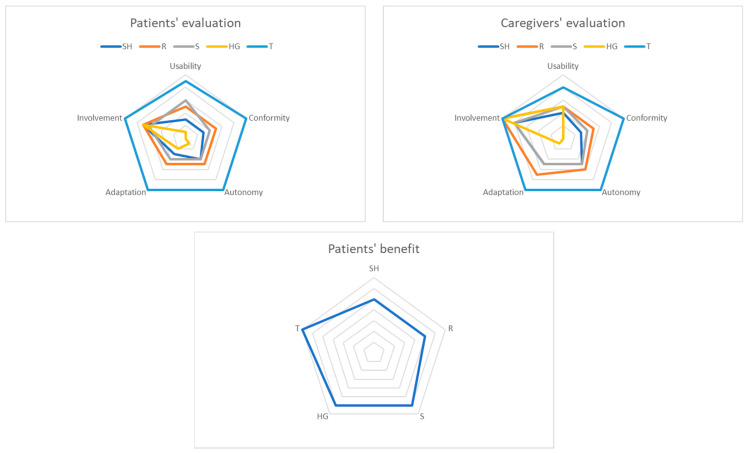
Radar representation of patients’ and caregivers’ assessments of the techniques. SH: self-hypnosis; R: relaxation; S: sophrology; HG: holistic gymnastics; T: transcutaneous electric nerve stimulation.

**Table 1 jcm-12-01722-t001:** Patient characteristics, pain, and quality of life at entry and three months after transplantation.

			At Entry n = 59	Three Months after Transplantation n = 59	*p*-Value
Age (years)			34 (28)		
Sex female/male			34 (57.6)/ 25 (42.4)		
Lung disease					
	Cystic fibrosis		32 (54.2)		
	Emphysema/COPD		16 (27.2)		
	Lung fibrosis		9 (15.2)		
	Other		2 (3.4)		
Pain					
	Presence of pain		23 (40.3) {2}	21 (35.6)	0.532
	Intensity of pain, VAS *				
		During the consultation	0 (18) {3}	0 (12)	0.630
		Maximum pain intensity during the last three days	16 (52) {3}	27 (50)	0.654
		Usual pain intensity during the last three days	6 (32) {3}	4 (25)	0.320
	Localization of pain				
		Thoracic pain	13 (30.9) {17}	18 (38.3) {12}	0.439
		Care related pain	8 (18.6) {16}	16 (34.0) {12}	0.098
	Neuropathic pain **		2 (3.4) {1}	16 (27.6) {1}	0.0008
	Analgesics				
		None	52 (88.1)	39 (66.1)	0.004
		WHO classification ***			0.088
		Class 1	2 (3.4)	9 (15.2)	
		Class 2	3 (5.1)	9 (15.2)	
		Class 3	2 (3.4)	2 (3.4)	
	Treatment of neuropathic pain		1 (1.7)	4 (6.8)	0.180
	Non-medicinal treatment		1 (1.7)	0 (0)	NA
	Benzodiazepine		7 (11.9)	11 (18.6)	0.206
Quality of life					
	HAD score ****				
		A	8 (8) {2}	5 (4)	<0.0001
		D	6 (6) {2}	2 (3)	<0.0001
	Perceived Stress Score		28 (12) {2}	22 (9)	<0.0001
	Spiegel score		18 (4) {2}	18 (6)	0.733
	EuroQol visual analogic scale		40 (33) {2}	75 (20)	<0.0001
	Ringsted score total			5 (8)	
		Routine		2 (5)	
		Task		2 (5)	

COPD: Chronic obstructive pulmonary disease. *: evaluated using a visual analogic scale (VAS) from 0 (=no pain) to 100 (=maximum imaginable pain intensity). **: Neuropathic pain is considered when DN4 score is ≥4 [18]. ***: World Health Organization (WHO) classification. ****: Hospital Anxiety and Depression Scale with its sub-scales for anxiety (A) and depression (D) Results are presented as median (interquartile range) and number (percentage). The number of missing data is presented as { }.

**Table 2 jcm-12-01722-t002:** Number of sessions performed by the whole group.

		Number of Sessions Performed		Total Number of Sessions
		Before Transplantation	After Transplantation	
Holistic gymnastics				
	Training Session	57	30	87
	Self-Practice	78	127	205
	Total	135	157	292
Self-hypnosis				
	Training Session	59	51	110
	Self-Practice	82	575	657
	Total	141	626	767
Relaxation				
	Training Session	54	23	77
	Self-Practice	613	1332	1945
	Total	667	1355	2022
Sophrology				
	Training Session	8	1	9
	Self-Practice	28	24	52
	Total	36	25	61
Transcutaneous electrical nerve stimulation (TENS)				
	Training Session	10	54	64
	Self-Practice	53	1100	1153
	Total	63	1154	1217

**Table 3 jcm-12-01722-t003:** Patients’ and caregivers’ assessment of the techniques.

		SH n = 50	R n = 51	S n = 3	HG n = 49	T n = 36	P Global	P between Techniques			
Patients											
	Usability	3 (7)	5 (10)	6 (8)	1 (6)	9 (4)	<0.0001	SH vs. T 0.0001	R vs. T 0.024	HG vs. T <0.0001	
	Conformity	3 (6)	5 (7)	4 (9)	0 (6)	10 (3)	<0.0001	SH vs. T <0.0001	R vs. T <0.0001	HG vs. T <0.0001	
	Autonomy	4 (7)	5 (10)	4 (8)	1 (6)	10 (2)	<0.0001	SH vs. T <0.0001	R vs. T 0.0003	HG vs. T <0.0001	R vs. HG 0.031
	Adaptation	3 (8)	5 (10)	4 (9)	2 (7)	10 (4)	<0.0001	SH vs. T <0.0001	R vs. T 0.0006	HG vs. T <0.0001	
	Involvement	7 (5)	7 (7)	6 (9)	7 (5)	10 (3)	0.0299	SH vs. T 0.0228			
	Benefit	5 (10)	5 (10)	6 (8)	6 (10)	7 (7)	0.3541				
Caregivers											
	Usability	4 (10)	5 (10)	5 (10)	5 (8)	8 (5)	0.0065	SH vs. T 0.025	HG vs. T 0.003		
	Conformity	3 (10)	5 (8)	4 (10)	0 (5)	10 (5)	<0.0001	SH vs. T 0.0001	R vs. T 0.003	HG vs. T <0.0001	R vs. GH 0.007
	Autonomy	5 (9)	6 (10)	5 (10)	0 (5)	10 (2)	<0.0001	SH vs. T 0.0002	R vs. T 0.007	HG vs. T <0.0001	R vs. GH 0.007
	Adaptation	5 (10)	7 (10)	5 (10)	1 (7)	10 (5)	<0.0001	SH vs. T 0.0431	HG vs. T <0.0001	R vs. GH 0.032	
	Involvement	8 (5)	10 (3)	8 (5)	10 (3)	10 (0)	0.0059	SH vs. T 0.0020			

SH: Hypnosis; R: Relaxation; S: Sophrology; HG: Holistic gymnastics; T: Transcutaneous electrical nerve stimulation. Results are presented as median (interquartile range) Qualities of the techniques, patient implication and satisfaction degree were measured by an 11-point numeric rating scale with 0 being “worst possible” and 10 “excellent” on five features (usability, conformity, autonomy, adaptation, involvement.

## Data Availability

Our dataset is available from the Dryad repository (DOI: 10.5061/dryad.tb2rbp03h).

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
