# Peer review of "Complementary Therapy Learning in the Setting of Lung Transplantation: A Single-Center Observational Study of Appropriation and Efficacy"

_jcm, 2023, doi:10.3390/jcm12051722_

Round 1

Reviewer 1 Report (New Reviewer)

Dear Authors

Thank you for the opportunity to review this manuscript. This approach is novel and has minimal content in the literature regarding this topic. 

1. Introduction: I would like to see more background information regarding the techniques in other transplant fields. 

2. The methods section is very long. I believe some of it should be placed in some form of a supplemental table/ figure

3. Was there an alternative to joining this? Were patients who did not want to participate looked at? Was there a comparison?

4. I see that one of the major goals were not achieved which was to alleviate pain. Why do you think this did not happen

Author Response

Reviewer 1

Thank you for the opportunity to review this manuscript. This approach is novel and has minimal content in the literature regarding this topic. 

Comment 1. Introduction: I would like to see more background information regarding the techniques in other transplant fields. 

Response to Comment 1. Our text included the sentence: “Mind-body techniques like relaxation, mindfulness and hypnosis are helpful for pain and anxiety management in perioperative care [10,11].” Reference 10 was about transplants. We have modified our text to be clearer (Line 47): “Mind-body techniques, like relaxation, mindfulness, and hypnosis, are used for pain and anxiety management in perioperative care. However, the quality of evidence for its efficacy for improving post-surgical outcomes is limited as reported in a systematic review [10]. Their place in solid organ transplantation has begun to be studied with favorable results after kidney, kidney/pancreas, liver, heart, or lung transplanted patients [11]. They have been more recently proposed for patients waiting for kidney transplantation [12,13]. In patients awaiting lung transplantation, higher levels of coping and lower levels of disengagement have been shown to be associated with better psychological quality of life [9]. Interestingly, telephone-based coping skills training allowed significant improvements in quality of life in these patients [14].”

Comment 2. The methods section is very long. I believe some of it should be placed in some form of a supplemental table/ figure

Response to Comment 2. We have modified our text by reducing paragraph 2.4. Study Protocol - Techniques. The description of the techniques is reported in a supplementary file.

 The revised text is (Line 89): “2.4. Study Protocol - Techniques

The opportunity to learn each technique was proposed to the participants. The global process of the study was explained to the patients, including the description of the different techniques, their usefulness, and the necessity to practice them regularly before and after the transplantation. A booklet recapitulated the information.

A description of the various techniques (relaxation, self-hypnosis, sophrology, and holistic gymnastics) is presented in a Supplementary File S1 (Study Protocol - Techniques). In addition, the possibility of using transcutaneous electric nerve stimulation (TENS) was mentioned to each patient but it was taught before transplantation only to patients describing a pain where TENS could be useful. TENS was taught after transplantation to each patient presenting a pain poorly relieved by epidural or usual analgesics. The technique was taught by pain nurses in charge of the study.

Repeated training sessions were possible pre transplantation in patients staying for a long time or repeatedly in the hospital before the transplantation, after the transplant, additional sessions were performed at the request of the patients and/or the team. This disparity is possibly a limiting factor in this study.”

The Supplementary File S1 (Study Protocol - Techniques) is:

“Relaxation was taught with the help of a smartphone application (RespiRelax). During the training session the patients were asked to breathe regularly for five minutes while silently counting (three seconds inhaling and three seconds exhaling), following a bubble on the smartphone moving up and down. While they were following the bubble, they were asked to relax the shoulders, then the jaws and then all the body through a body scan. Patients were asked to practice this exercise three times a day, for five minutes. They were told that the aim of this exercise was to increase their muscular relaxation, to decrease stress, to decrease the sympathetic tonus, and to favor the parasympathetic state. They could also use it to focus their attention during an uncomfortable procedure. The technique was taught by two psychologists and two anesthesiologists in the same way.

The self-hypnosis group was taught during a 20-minute session. The same type of induction and therapeutic suggestions was used by the investigators without a script. Induction was based on regular respiration, focus on surrounding sounds, specific areas of the body, and therapeutic suggestions were based on the patient's own memories. The patients were taught to visualize themselves in a safe place, or doing their favorite activity, they could learn how to increase their comfort and to “protect” themselves in an uncomfortable situation. For example, the metaphor of a magic glove protecting the hand and the wrist was taught to increase the comfort during blood gas sampling. At the end of the session, patients received a CD or a link(https://www.dropbox.com/s/ax2hsupg8e56bnf/fauteuil%20de%20nuages.wav?dl=0) with relaxation and self-hypnosis exercises (SACEM 215517/2013; https://societe.sacem.fr/en). The patients were told to practice regularly at home, they would use the technique to increase their comfort during an uncomfortable or painful procedure or during a stressful situation (i.e., endoscopic examinations, uncomfortable care, sleep difficulties…). The technique was taught by two psychologists and two anesthesiologists. All of them had done the same academic 1-year hypnosis course at the Paris-Saclay University. They all had more than 8 years’ experience at the time of the study. To avoid any discrepancy, they ascertained regularly that they had the same practice.

The sophrology exercise was taught during a 30-minute session with practical physical and mental exercises, using techniques such as concentration, deep breathing, relaxation, visualization, and simple movements. The patients were told to practice regularly at home and to use the techniques as necessary to increase their comfort during an uncomfortable or painful procedure or a stressful situation. The technique was taught by a registered nurse who had 4 years’ training and had a master’s degree in “sophrologie caycedienne” (https://sofrocay.com/). She recorded the session on the cell phone of the patient for ulterior training.

The holistic gymnastics session consists of learning three short exercises of simple corporal movements, stretching and abdominal breathing. The technique was taught by a physiotherapist who had done a 2-year course in holistic gymnastics. Patients were told to practice these exercises as routine training every day. “

Comment 3. Was there an alternative to joining this? Were patients who did not want to participate looked at? Was there a comparison?

Response to Comment 3. As shown in Figure 2 (flow chart), 84 patients were assessed for eligibility and only 4 did not agree to participate. Consequently, the question of comparing the included patients with these patients did not arise.

Comment 4. I see that one of the major goals were not achieved which was to alleviate pain. Why do you think this did not happen

Response to Comment 4. It’s true and we have added the following sentences at the end of the Discussion section (Line 325): “Finally, pain management after lung transplantation is complex. In addition to the fact that double-lung transplantation involves two thoracotomies, with their own risk of chronic pain, patients also suffer from pain related to care and to various postoperative complications,... The functional result also plays a role when it is not "perfect" influencing the psychological state.”

Reviewer 2 Report (New Reviewer)

Well written paper. Statistics is correct.

Well written paper under the methodological and statistical point of view.

English as well is correct.

Strong point of the study is the investigation of complementary therapy in the context of lung transplant which has never been explored so deeply in the past.

Weak points of the study are the small number of patients putting into practice all the techniques proposed and that there was a wide variety of techniques, making it difficult to evaluate  every single technique.

Author Response

Reviewer 2

Well written paper. Statistics is correct.

Well written paper under the methodological and statistical point of view.

English as well is correct.

Strong point of the study is the investigation of complementary therapy in the context of lung transplant which has never been explored so deeply in the past.

Weak points of the study are the small number of patients putting into practice all the techniques proposed and that there was a wide variety of techniques, making it difficult to evaluate  every single technique.

Response. The reviewer points out the small number of patients. This is specified in paragraph 4.1. Weakness and strength of the study: “First, the main weakness of the study is the small number of patients able to learn all the techniques. Only 75% of the patients had self-hypnosis and four patients had sophrology. It was due mainly to the variable availability of the facilitators. Consequently, external validity is limited.”

Reviewer 3 Report (New Reviewer)

This is a novel prospective study looking into feasibility of implementation and self-appropriation of  complementary techniques in lung transplantation patients. It failed to show any statistical improvement in the symptoms but with further work and improvement in the techniques- this can be made standard of care in all solid organ transplant cases. 

Author Response

Reviewer 3

This is a novel prospective study looking into feasibility of implementation and self-appropriation of  complementary techniques in lung transplantation patients. It failed to show any statistical improvement in the symptoms but with further work and improvement in the techniques- this can be made standard of care in all solid organ transplant cases. 

Response. We thank the Reviewer for this comment.

This manuscript is a resubmission of an earlier submission. The following is a list of the peer review reports and author responses from that submission.

Round 1

Reviewer 1 Report

I read with interest this study on complementary therapies in the setting of lung transplantation. I believe indeed that there could be a place for them for lung transplant candidates, since these patients very often come with a huge load of anxiety, fear and depression. The time which passes from listing to transplantation offers an ideal window of time wherein the patient can be taught and can practice such therapies.

I'm a bit perplexed about the study design. These are very different therapies from each other, which have also different purposes. Proposing 4/5 different treatment to a patient could be confusing for him and it could be difficult to understand the effect of the treatment itself on any patient reported outcome (which I understand it was not the main purpose here), or in any case it could influence the ability of the patient itself to learn more than one technique at the same time (which was the purpose of the study).

Moreover, the choice of the treatments to propose seemed to be dictated more by the local availability of a qualified teacher in that specific treatment than by more rational criteria.

So, as the authors pointed out, I think there's a problem with external validity of the study.

About the paper itself, it is well presented, I have these comments:

1) Is it really necessary to put both interquartile range and 95% CI for every variable in the tables? They become really crowded with numbers and I think the authors should try to improve them. 

2) Some patients had repeated training sessions. Both before and after lung transplantation. Why so? This happened when the patient didnt feel comfortable with that technique? Was that scheduled or only on patient's request? Could you add a comment about that? 

3) In table 2, number of sessions performed is not so much meaningful in my view if you do not specify how many sessions for each single patient.

4) I'm not quite sure what the "investment" parameter actually measured. Could you provide a brief description?

5) About the radar plots. It is evident that TENS is very easy to learn and useful. And this is my point, it is too different from other treatments and I'm not sure it should be compared with them

Reviewer 2 Report

I congratulate you on writing the results of your study, I’ve some comments.

Comment 1. I believe the authors should improve the presentation of the figures:  both figure 1 and 3 are a screenshot and there are still the red underlinings, please correct it.

Comment 2. For a clinician without expertise in these practices the fact that they don't improve pain, anxiety  and quality of life, might be more important than their feasibility.

Round 2

Reviewer 2 Report

I suggested to reject the manuscript and even if the Authors made some changes (that I carefully read) I still think this paper should be rejected.